# The structure and catalytic mechanism of a pseudoknot-containing hammerhead ribozyme

Xuelin Zhan[1,2,3,6], Timothy J. Wilson [4,6], Zhenzhen Li[1,2], Jingjing Zhang[1,2], Yili Yang[3], David M. J. Lilley [4,5] ✉ & Yijin Liu[1,2] ✉

We have determined the crystal structure of a pseudoknot (PK)-containing hammerhead ribozyme that closely resembles the pistol ribozyme, with essentially identical secondary structure and connectivity. The activity is more sensitive to deletion of the G8 2'OH than to the absence of magnesium ions, indicating that the catalytic mechanism is the same as the extended hammerhead, and distinct from the pistol ribozyme. Here we show that nucleophilic attack is almost perfectly in-line, and the G8 2'OH is well positioned to act as general acid, being directed towards the O5' leaving group, and 2.9 Å away from it. Despite the similarity in overall structure to the pistol ribozyme, the local structure close to the cleavage site differs, and the PK hammerhead retains its unique mechanistic identity and demonstrates enhanced activity over other hammerhead ribozymes under standard conditions.

RNA enzymes (ribozymes) are RNA molecules that catalyse chemical reactions. Ribozymes likely played an important role in the chemical origins of life on the planet, but remain important in contemporary biology, catalysing important reactions in the cell. In addition they have attracted interest as potential therapeutic agents. Most known ribozymes catalyse phosphoryl transfer reactions, many of which are classified as nucleolytic ribozymes. The hammerhead ribozyme is a widespread example[1,2], including within the human genome[3–5], and was one of the first to be discovered[6–8].

The core of the hammerhead ribozyme comprises a three-way helical junction (Fig. 1A) containing largely conserved nucleotides of which G12 and G8 are functionally important. The helices are conventionally labelled I, II and III[9]. Crystal structures were determined for this minimal form of the ribozyme[10,11], yet discrepancies were found between the structure and biochemical analysis of the cleavage reaction[12]. It was later discovered that the natural form included longer-range tertiary interactions that stabilised the ribozyme[13–16] (Fig. 1B). These forms are often termed extended hammerhead ribozymes. The long-range interactions comprised loop-loop (internal or

terminal) interactions between helices II and I. Crystal structures were determined for two forms by Scott and co-workers, with tertiary interactions between two terminal loops[17] and between a terminal loop of helix II and an internal loop in helix I[18] (Fig. 1B). Relatively recently Breaker and co-workers[19] used bioinformatics to identify a new form of hammerhead ribozyme in which a loop corresponding to helix II forms a pseudoknot with an additional stem-loop connected to helix I (Fig. 1C). Interestingly this secondary structure is closely similar to that of the pistol ribozyme[19,20] (Fig. 1D). There have been a number of crystallographic studies of the pistol ribozyme[21–23].

As a group the nucleolytic ribozymes all employ general acid-base catalysis for the cleavage and ligation of phosphodiester linkages. They can use nucleobases (especially guanine), 2'-hydroxyl groups and hydrated divalent metal ions as functional groups for this purpose, and the group can be subdivided according to which are employed in a given ribozyme[24]. Many of these use guanine N1 as the general base in the cleavage reaction in order to deprotonate the O2' nucleophile, and this is the case for both the hammerhead[25,26] and pistol[21,23] ribozymes. However, the two sub-classes of ribozyme differ in the nature of the

[1]State Key Laboratory of Medicinal Chemical Biology, Frontiers Science Center for Cell Responses, Nankai University, Tianjin, China. [2]College of Pharmacy, Nankai University, Tianjin, China. [3]China Regional Research Centre, International Centre of Genetic Engineering and Biotechnology, Taizhou, P. R. China. [4]Division of Molecular, Cellular and Developmental Biology, The University of Dundee, Dundee, UK. [5]Visiting Professor, Nankai University, Tianjin, China. [6]These authors contributed equally: Xuelin Zhan, Timothy J. Wilson. ✉e-mail: d.m.j.lilley@dundee.ac.uk; yvliu@nankai.edu.cn

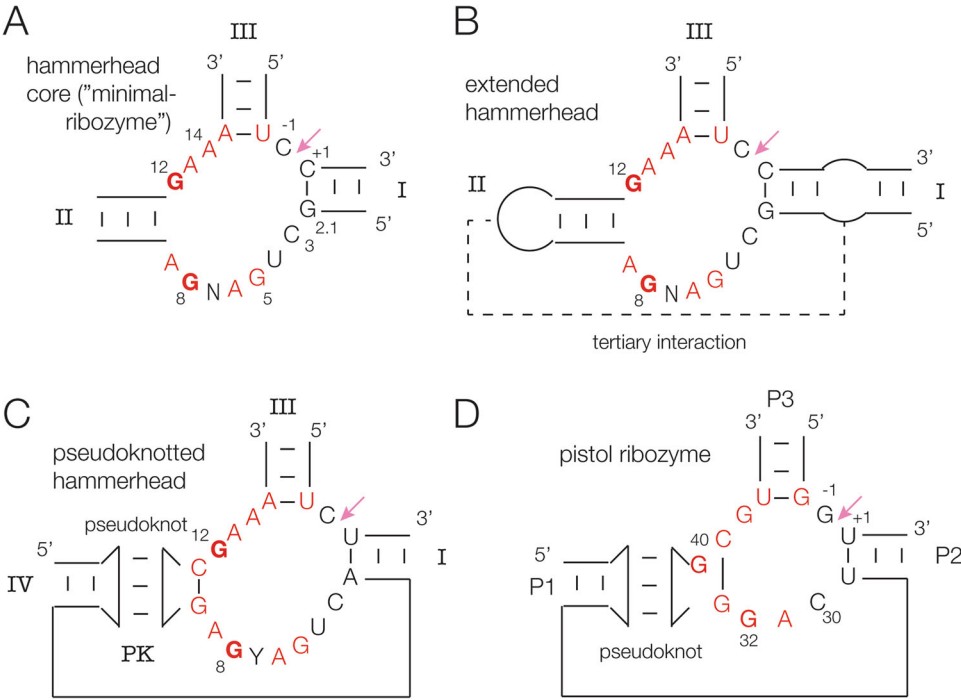

**Fig. 1 | Secondary structure schemes and core sequences of hammerhead and pistol ribozymes. In each case the cleavage site is arrowed, and key nucleotides labelled. A** The core of the minimal hammerhead ribozyme structure. The helices and core nucleotides are labelled according to the standard nomenclature[9]. This has also been applied to the other hammerhead forms. **B** The extended hammerhead ribozyme[13,14]. The loop-receptor interaction is indicated by the broken line.

**C** The pseudoknot-containing hammerhead ribozyme[19]. We have termed the pseudoknot helix PK, and it's stem helix IV. **D** The pistol ribozyme[19,20]. The nomenclature for the helices is different for this ribozyme. There is no standard nomenclature for the nucleotide positions, so we have used those in the PDB file (ID 6R47) for this ribozyme.

general acid that protonates the O5' leaving group in the cleavage reaction. Two alternative mechanisms have been described[23], in which the general acid is:

1. A 2'-hydroxyl activated by an adjacent divalent metal ion, or
2. an inner sphere water molecule of a metal ion coordinated by an adjacent 2'-hydroxyl group.

In the extended hammerhead ribozyme the general acid is the 2'-O of G8 (probably activated by a nearby divalent metal ion) i.e. mechanism (1). This is consistent with 5'-phosphorothiolate modification studies[27]. By contrast in the pistol ribozyme an inner-sphere water molecule of a divalent metal ion (coordinated in part by a ribose O2') acts as the general acid i.e. mechanism (2). The O2'−O5' distance was found to be 3.2 Å in the extended hammerhead structure[18], consistent with the role of G8 O2' as general acid. We found that the corresponding distance in the pistol ribozyme was 4.7 Å, while an inner sphere water molecule bound to a magnesium ion present in the active center was directly bonded to the O5' atom[23]. These two mechanisms were distinguished by the pH dependence of cleavage rates when the hammerhead G8 O2' and its equivalent in pistol were replaced by an amino group[23].

Given the close similarity in secondary structures between the pseudoknot-containing form of the hammerhead and the pistol ribozymes, a number of questions arise. Does the similarity extend to the three-dimensional structures? And does the mechanism of the pseudoknot-containing hammerhead conform to that of the extended hammerhead ribozyme (mechanism 1), or does it adopt the same mechanism as the pistol ribozyme (mechanism 2)? In this work, we have therefore solved a crystal structure of the pseudoknot-containing hammerhead in its pre-reaction state. The global structure is similar to that of the pistol ribozyme, but the cleavage reaction likely proceeds via mechanism (1) in common with other forms of the hammerhead ribozyme.

## Results

### RNA preparation, crystallization, and structure determination

A pseudoknot-containing hammerhead ribozyme (PK hammerhead) sequence was designed to recognize a target sequence in RNA; sequences are listed in Materials and Methods. The ribozyme strand was synthesized by in vitro transcription by T7 RNA polymerase, while the substrate strand was chemically synthesized to include a 2'H substitution at the cytosine 5' adjacent to the scissile phosphate to prevent cleavage during crystallization. A derivative containing a 5-bromocytosine substitution in the substrate strand was synthesized in order to provide phase information. The hybridized molecules crystallized in the hexagonal space group $P6_122$. Crystallographic data were acquired from the native and derivative crystals, at resolutions of 2.89 Å and 3.5 Å, respectively (see Supplementary Table 1). After the acquisition of phase information from the derivative dataset through a single-wavelength anomalous diffraction signal, the native datasets were employed for refinement and modeling. The coordinates of the PK hammerhead model are deposited in the PDB with ID 8YDC.

### The overall structure of the pseudoknot-containing hammerhead ribozyme

The overall structure of PK hammerhead is shown in Fig. 2, and the electron density map is shown in Supplementary Fig. S1. Where possible we have used the standard nomenclature for the hammerhead ribozyme core nucleotides[9], except that we have named the nucleotides immediately 5' and 3' to the scissile phosphate C-1 and U +1 respectively. Supplementary Table S2 shows the correspondence between these names and the numbering in the PDB file. The structure is distinct from previously reported forms of hammerhead ribozyme structures in that it includes a pseudoknot in place of other tertiary contacts. We have named the helices to be

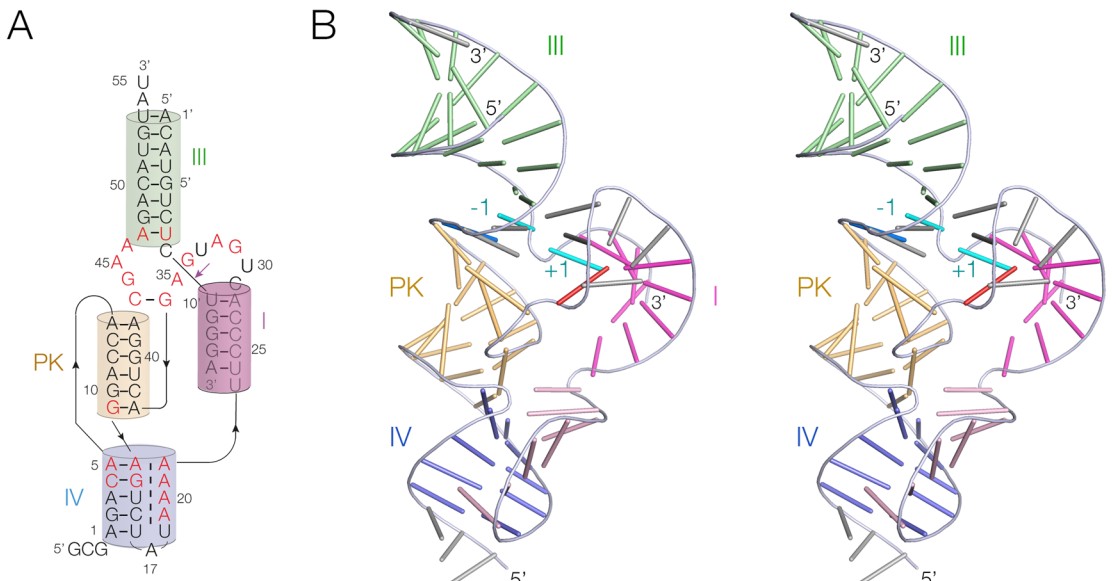

**Fig. 2 | The overall structure of the pseudoknot-containing hammerhead ribozyme. A** Schematic of the secondary structure of the PK hammerhead ribozyme. Highly-conserved nucleotides are written in red. The coloring of the helices matches those in the images of the 3D structure. **B** Parallel-eye stereoscopic view of the 3D structure of the PK hammerhead ribozyme observed in the crystal structure. For clarity in viewing the overall structure, each nucleotide is depicted as a bar in this image. The coloring of the helices matches the schematic shown in (**A**). The −1 and +1 nucleotides flanking the scissile phosphate are colored cyan.

consistent with the standard naming in the hammerhead ribozyme[9]. Helices I and III have the same role as in the standard hammerhead structure, while helix II has been replaced by the pseudoknot interaction. The pseudoknot (PK) is formed from a stem loop (we term helix IV), and is a 6 bp helix comprising four standard *cis*-Watson Crick base pairs, capped at either end by A:A and G:A *cis*-Watson Crick base pairs (Fig. 3A). The PK helix essentially fulfils the same role in the global structure that helix II plays in the standard hammerhead ribozyme structure. The global structures of the PK and extended forms of the hammerhead ribozyme are compared in Supplementary Fig. S2.

The stem of the loop forming the PK helix (helix IV) comprises five Watson-Crick base pairs, but forms a minor-groove triplex interaction with the AUA$_4$ sequence that connects helix IV to helix I (Fig. 3B). The third stand nucleobases are mutually stacked, and inclined at ~40° to the base pair planes of helix IV, and form hydrogen bonding interactions with the nucleobase heteroatoms and ribose O2' atoms in standard A-minor interactions[28] in most cases (Fig. 3C, D). Similar minor-groove triple interactions have been observed in other RNA structures including the pistol ribozyme[21,23] and the NAD$^+$-II riboswitch[29].

### Structure of core of the pseudoknot-containing hammerhead ribozyme

The core of the PK hammerhead ribozyme lies at the junction of helices I, PK and III, and comprises eleven highly conserved nucleotides (Fig. 1C). C3 through A9 (nucleotides 29 to 35 in the PDB file – see Supplementary Table S2) loop around 180° with a sharp turn between U4 and G5 (30, 31 respectively in PDB) (Fig. 4A). This is very similar to the standard hammerhead conformation; the loops of the PK and extended forms of the hammerhead are closely similar (Supplementary Fig. S3), and the loops of the sTRSV- and the PK hammerhead ribozymes can be superimposed with an RMSD of 0.292 Å. C3 is stacked upon A2.1 at the end of helix I, and on the distal side of the turn G5 and A6 are also mutually stacked. At U7 the backbone rotates into the interior of the turn, such that the nucleobase of U7 stacks upon C3, and G8 stacks upon U + 1 and base pairs with C3. There is another sharp turn between G8 and A9, and A9 forms a sheared base pair with G12 to

form the short stem of the loop that forms the PK helix along with a standard base pair between G10.1 and C11.1.

### Structure of the active center of the pseudoknot-containing hammerhead ribozyme

The ribozyme cleavage site is located between C-1 and U + 1 (nucleotides 9 and 10 respectively in the PDB file). The phosphate backbone of the substrate strand forms a sharp turn close to 180° at the scissile phosphate, generating the 'splayed apart' conformation required to give an in-line attack geometry. The RNA that was crystalized lacked the 2'-hydroxyl group at C-1 (in order to prevent ribozyme cleavage), and we have therefore modeled this in place (Fig. 5A), switching the ribose pucker to C3'-*endo* conformation. The structure was then re-refined against the crystallographic data. The resulting conformation is close to perfect for in-line nucleophilic attack by the C-1 O2', with an O2' – P distance of 3.0 Å and a dihedral angle of 172.5°. This corresponds to an in-line fitness parameter[30] $F = 0.94$. The conformation around the scissile phosphate is stabilised by nucleobase interactions on both sides. The nucleobase of C-1 (5' to the scissile phosphate) is stacked upon that of G12. U + 1 is base paired with A2.1 at the top of helix I, and its nucleobase is stacked between those of G8 and A1.1. This conformation is closely similar to that of the extended hammerhead ribozyme[18].

The conformation around the cleavage site (Fig. 5A) is fully consistent with the nucleobase-mediated general acid-base catalysis observed in other hammerhead species. The general base in the hammerhead ribozyme has been deduced to be G12[25,27], and in the PK hammerhead ribozyme G12 N1 is 2.7 Å from the O2' nucleophile in the cleavage reaction, so that it is well placed to remove the proton from the nucleophile. The O2' of G8 has been proposed as the general acid in the extended hammerhead ribozyme[18,23,31], and the G8 O2' is 2.9 Å from the O5' leaving group in the PK hammerhead ribozyme, so that it is well positioned to donate a proton to the leaving group in the cleavage reaction. Thus, like the extended hammerhead ribozyme, G12 and G8 are in position to act as general base and acid to mediate the required proton transfers. By contrast, the corresponding view of the pistol ribozyme (Fig. 5B) shows the different orientation of G32 leading to the longer O2-O5' distance.

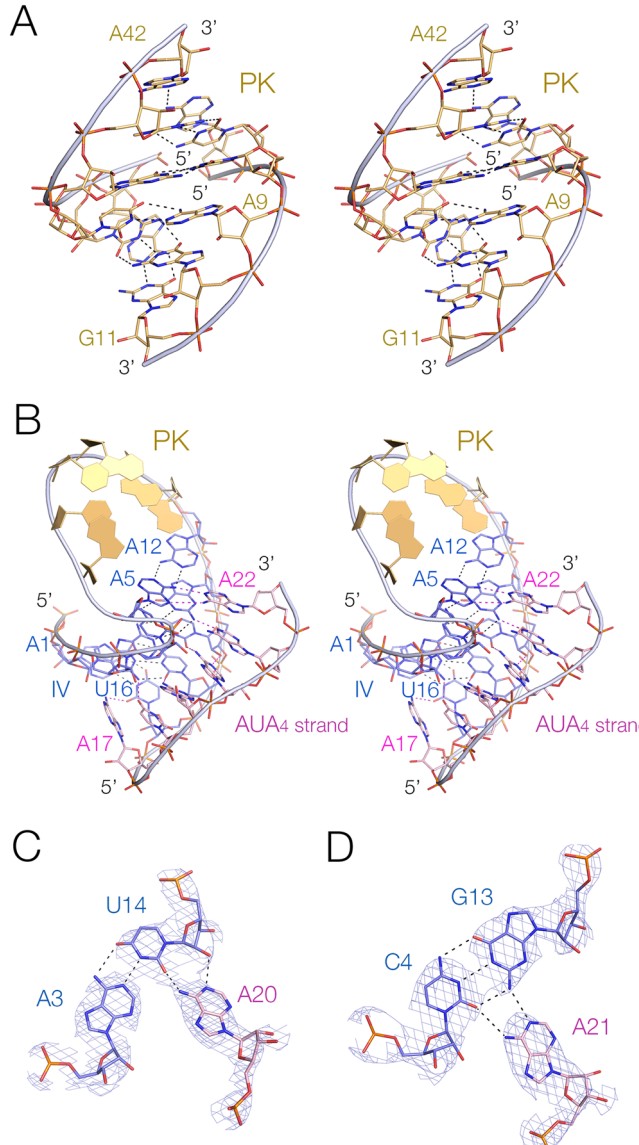

**Fig. 3 | The pseudoknot helix and the minor-groove triple helix in the PK hammerhead ribozyme. Parts A and B are shown in parallel-eye stereoscopic views. A** The pseudoknot helix. This comprises a 6 bp standard A-form RNA helix, with non-Watson-Crick base pairs at each end. **B** Helix IV, the stem of the pseudoknot helix. This forms a triple helix by binding the AUA₄ sequence in its minor groove. Representative base triples in the IV triplex formed by A20 (**C**) and A21 (**D**). The electron density maps are 2F$_o$-F$_c$ maps contoured at 2.5 σ.

## Comparison of the structures of the PK hammerhead and pistol ribozymes

Comparison of the 2D and 3D structures of the new PK hammerhead ribozyme with that of *Aliistipes putredinis* genome (*ap*Pistol, PDB: 6R47) pistol ribozyme[23] shows that both ribozymes contain a pseudoknot (PK) of six base pairs, indicating significant similarities in their three-dimensional structures. In both ribozymes the central helix (PK in the PK hammerhead and PS in pistol) is coaxial with helix III (P3 in pistol) and helix IV (P1 in pistol), while P1 (P2 in pistol, where it is 1 bp shorter) is adjacent and parallel to PK. The connectivity of these helices is identical (Supplementary Fig. 4) and both have an A-minor triple interaction with helix IV (P1 in pistol). Thus the overall fold of the ribozymes are closely similar (Supplementary Fig. 5A, B), demonstrated by superpositions of the structures in Supplementary Fig. 5C.

The main difference between the PK hammerhead and pistol ribozyme structure lies in the loop that connects helix I of the hammerhead (P2 in pistol) and the pseudoknot helices in each ribozyme, shown in Fig. 4. In both ribozymes the backbone of the loops turn by approximately 180°, but the hammerhead loop is significantly longer than that of the pistol ribozyme (Fig. 4 A, B). In the hammerhead ribozyme the loop has the sequence 5′ C3 U / G A6 U G̲ A9 3′, whereas in the pistol ribozyme it is 5′ C30 / A G̲ A33 3′. The G8 nucleotide in the hammerhead proposed to act as general acid and it's equivalent in the pistol ribozyme (G32) are underlined. The position of the sharp turn in the backbone is indicated by a slash (/) in each sequence.

The loop of the pistol ribozyme is relatively simple. U29 and C30 are stacked with the helical geometry of P2 before the turn. U29 hydrogen bonds with U + 1 5′ to the scissile phosphate, helping to achieve the in-line attack analogous to that of the hammerhead ribozyme discussed above. C30 turns into the center of the loop and stacks on the U + 1:U29 base pair, after which A31 and G32 are mutually stacked and directed away from the minor groove side of the loop. Importantly, G32 N7 accepts a hydrogen bond from the O2′ of C30, holding it in place. The loop of the hammerhead ribozyme is a little more complex, as described in the previous section. After the turn at U4, G5 and A6 are stacked together and directed away from the minor groove face of the loop. U7 then turns into the center of the loop and stacks on the nucleobase of C3. Importantly, G8 is also directed into the center, and base pairs with C3, so that the nucleobase of G8 is stacked on that of U + 1.

From a mechanistic perspective, the key difference between the PK hammerhead and pistol ribozymes is the local conformation of G8 (hammerhead) and G32 (pistol). In the PK hammerhead ribozyme the base pairing of G8 directs its ribose towards the reaction center such that its O2′ is 2.9 Å from the O5′ with a C-O2′-O5′ angle of 102°, so the proton is directed towards the leaving group oxygen atom at close to the tetrahedral angle. The position of the G8 nucleobase in the PK hammerhead structure is known with confidence, shown by the electron density map shown in Supplementary Fig. S6. In the pistol ribozyme G32 is not base paired, and rotated away from the loop so that the equivalent O2′-O5′ distance is increased to 4.6 Å, and the C-O2′-O5′ angle is rather less favorable for proton donation at 76°.

## Mechanistic analysis of the PK hammerhead and comparison with the pistol ribozyme

The measured rate of cleavage by the PK hammerhead ribozyme under single-turnover conditions in our standard buffer (50 mM TAPS (pH 8.0) 1.0 mM MgCl₂, 2 M NaCl, 0.1 mM EDTA) at 25 °C is 16 ± 1 min⁻¹ (Fig. 6; Table 1). This is an order of magnitude faster than the extended hammerhead ribozyme under the same conditions, and almost twice as fast as the pistol ribozyme[23]. It is therefore likely that the pseudoknot structure creates the optimal conformation in the active center. Our crystallographic data show that the O2′ nucleophile attacks in an almost perfect in-line conformation, and the G12 N1 and G8 O2′ are each at an optimal distance for the required proton transfers (Fig. 5).

We have previously shown that while both the extended hammerhead and pistol ribozymes use G N1 as general base in cleavage, they differ in the general acid. The hammerhead uses G8 O2′ to donate a proton to the O5′ oxyanion leaving group, while the pistol ribozyme uses a magnesium ion-bound water molecule in this role. The crystallographic data suggest that the PK hammerhead is likely to use G8 O2′ as general acid, but we have explored this using biochemical analysis. We have studied the rate of cleavage of the PK hammerhead in the absence of divalent cation (noting that a high concentration of monovalent ion is always present, i.e. 2 M NaCl), or with a dG8 (G8 O2′H) substitution (Fig. 6). The results are summarised in Table 1, where they are also compared with analogous data for the extended hammerhead and pistol ribozymes.

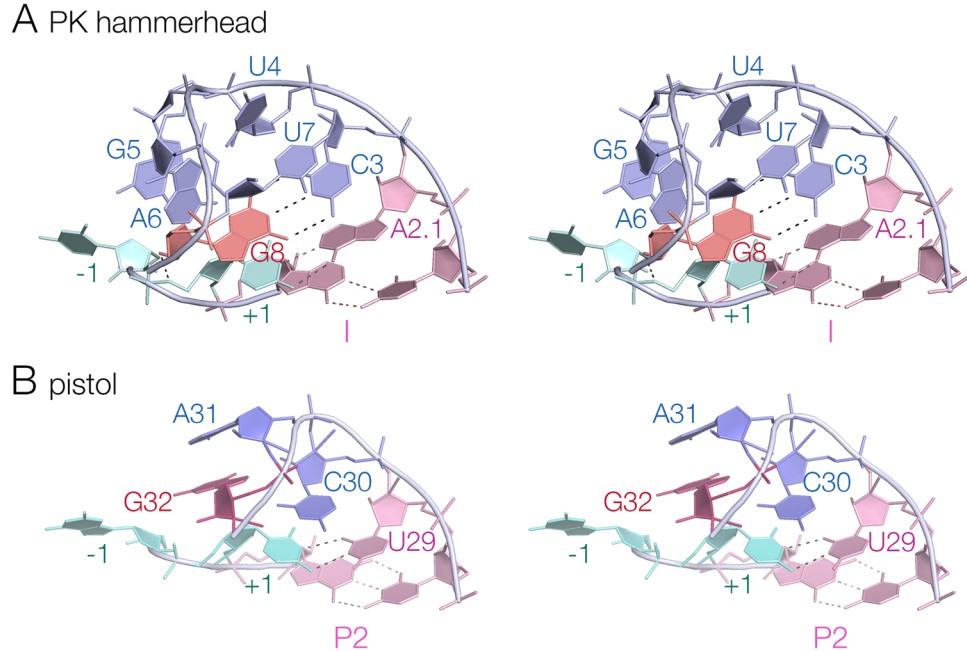

**Fig. 4 | The structures of the loop-turns in the core of the PK hammerhead and pistol ribozymes.** *Parallel-eye stereoscopic views are presented.* **A** The turn of the PK hammerhead ribozyme from A2.1 to G8. Note that G8 lies within the center of the loop, base paired to C3. **B** The turn of the pistol ribozyme from U29 to G32. In contrast to G8 in the PK hammerhead ribozyme structure, G32 is reoriented and directed away from the loop on the minor groove side. This image was generated from PDB ID 6R47[23].

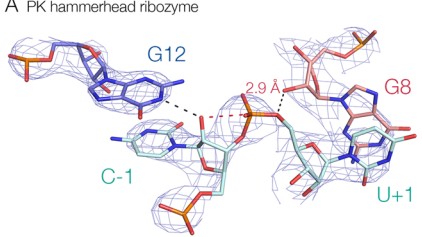 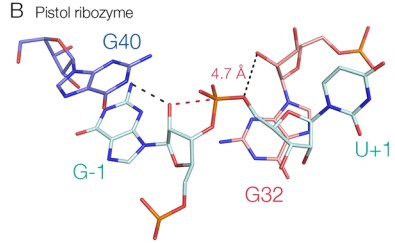

**Fig. 5 | The structure at the cleavage site of the PK hammerhead ribozyme, and comparison with the corresponding structure in the pistol ribozyme. A** The PK hammerhead structure. The red broken line indicates the direction of attack by the O2′ nucleophile. G12 is well placed to act as general base to remove the proton from the nucleophile, while G8 is positioned to act as general acid using its O2′ to protonate the O5′ leaving group, at a distance of 2.9 Å. The electron density is shown as the 2$F_o$-$F_c$ map contoured at 2.5 σ. **B** The pistol ribozyme structure. G32 is reoriented such that the O2′ - O5′ distance is 4.7 Å.

We find that in the absence of magnesium ions the rate of cleavage is only lowered 4.6-fold, whereas the loss of the 2′ OH of G8 lowers the rate 25-fold. When the dG8 variant was studied in the absence of magnesium ions (i.e. combining both changes) the rate was substantially lowered by 14,000 fold. Comparing these data with those from the extended hammerhead ribozyme we find similar effects. Absence of magnesium ions lowered the rate by 3.4-fold, while the dG8 substitution led to a 171-fold reduction in cleavage rate. Thus for both forms of the ribozyme the larger effect resulted from the loss of the G8 O2′ hydroxyl group. This is consistent with a primary role of the G8 O2′ as proton donor to the oxyanion leaving group in cleavage, also consistent with its observed position in the crystal structure. As with the PK hammerhead, the effect of simultaneous loss of magnesium ions and the dG8 substitution on the extended hammerhead is substantial (11,000 fold reduction).

For the pistol ribozyme, the absence of magnesium ions lowers the cleavage activity by 530 fold, while the loss of the corresponding O2′ led to a 66 fold reduction in cleavage rate. Thus in contrast to both hammerhead variants, the pistol ribozyme is more sensitive to the loss of the divalent metal ion than it is to loss of the O2′ group. Once again when the two changes are combined, the pistol ribozyme exhibits a large loss of activity (20,000 fold).

To summarise, rates of cleavage by the PK and extended hammerhead ribozyme are each more sensitive to loss of the G8 O2′ oxygen atom than to the absence of magnesium ions. By contrast this is reversed for the pistol ribozyme, where the absence of magnesium ions leads to a greater reduction in cleavage rate than the loss of the corresponding O2′ hydroxyl group. This indicates that while the pistol ribozyme employs an inner-sphere water of hydration of the bound magnesium ion observed in the crystal structure[21,23], both forms of the hammerhead use the O2′ hydroxyl of G8 as the general acid. Thus despite folding in a very similar manner to the pistol ribozyme, mechanistically the PK hammerhead remains very much a hammerhead ribozyme.

## Discussion

We have found that the pseudoknot-containing hammerhead ribozyme[19] is structurally very similar to the pistol ribozyme, both in terms of the connectivity of the helical sections and the three-dimensional structure. The pseudoknot forms are compact, and are likely to be very stable. We have found that the PK hammerhead is very

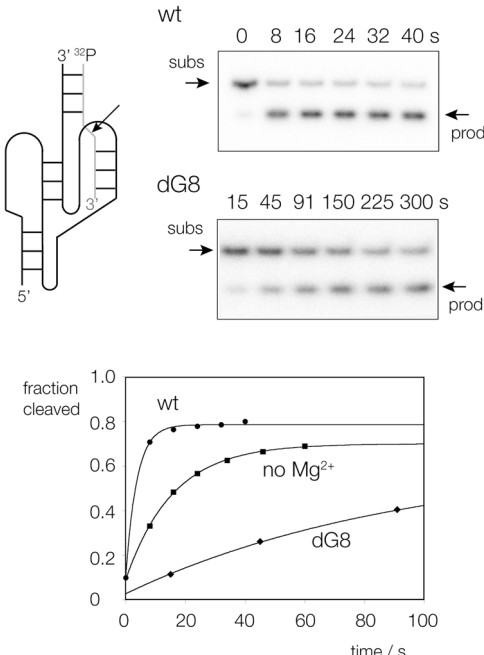

**Fig. 6 | The cleavage activity of the PK hammerhead ribozyme, and the effect of dG8 substitution and absence of magnesium ions.** A schematic of the construct (left) shows the two-piece ribozyme plus substrate (cleavage site arrowed), radio-actively [5′-$^{32}$P]-labelled at the 5′ terminus of the substrate. Ribozyme incubations are shown on the right for wild-type and dG8 PK hammerhead, showing separation of substrate and product by polyacrylamide gel electrophoresis under denaturing conditions as a function of time of incubation. Reaction progress is plotted as a function of time (lower) for wild-type (circles) and dG8 ribozyme (diamonds) in the presence of magnesium ions, and wild-type ribozyme in the absence of magnesium ions (squares). The data have been fitted to single exponential functions from which the rates of reaction have been calculated (Table 1). See also Supplementary Fig. S7 for uncropped gel image.

**Table 1 | Rates of cleavage by hammerhead and pistol ribozymes**

| Ribozyme | MgCl$_2$ + NaCl | | NaCl only | |
| --- | --- | --- | --- | --- |
| | Rate/min$^{-1}$ | Fold | Rate / min$^{-1}$ | fold |
| PK hammerhead | 16 ± 1 | (1) | 3.5 ± 0.3 | 4.6 |
| dG8 | 0.66 ± 0.02 | 25 | 0.0011 ± 0.0004 | 14,000 |
| Extend. hammerhead | 0.72 ± 0.03 | (1) | 0.21 ± 0.002 | 3.4 |
| dG8 | 0.0042 ± 0.0001 | 171 | (6.4 ± 0.2) x 10$^{-5}$ | 11,000 |
| Pistol | 9.77 ± 0.6 | (1) | 0.0184 ± 0.00003 | 530 |
| dA32 | 0.15 ± 0.1 | 66 | 0.0005 ± 0.00002 | 20,000 |

The rates for the PK hammerhead ribozyme were measured in the present work. Most of the data for the extended hammerhead and pistol ribozyme were taken from our earlier study[23] except for those for the pistol ribozyme in the absence of Mg$^{2+}$ that were measured in the present work. Note that all the measurements were made in a background concentration of 2 M NaCl. Fold reductions in rate are relative to ribozyme with a 2′-hydroxyl group at position 8 (hammerhead) or 32 (pistol), in the presence of 1 mM MgCl$_2$ and 2 M NaCl. Cleavage rates for the PK hammerhead ribozyme were determined in triplicate, and the mean and standard deviation reported.

ribozyme. The new crystal structure of the PK hammerhead is completely in agreement with this conclusion. While the overall architecture of the two ribozymes is essentially the same, the core is different (Fig. 4), and this results in a different position of G8 and G32 in the loops of the ribozymes so that the 2′-hydroxyl groups are positioned differently with respect to the O5′ leaving group in terms of distance and orientation. In the PK hammerhead the O2′ of G8 is directed towards the O5′ leaving group at a distance of 2.9 Å, whereas the corresponding distance in the pistol ribozyme was 4.6 Å, with a sub-optimal geometry.

One aspect of the kinetic data is worthy of further discussion. For both forms of the hammerhead ribozyme and the pistol ribozyme, the effect of simultaneously removing the O2′ and withholding magnesium ions on cleavage rate is much greater than either perturbation alone. In principle, both of the potential mechanisms will require the hydroxyl group and the divalent metal ion. For the hammerhead ribozyme the evidence indicates that the hydroxyl group is the general acid, but it is likely that it is activated by the proximity of a bound magnesium ion[31,35]. And for the pistol the metal ion that provides the bound water molecule that acts as the general acid is partially held in position by coordination by the G32 2′ hydroxyl group[23]. So in principle both ribozymes could respond to loss of both hydroxyl and divalent metal ion. That said, it is difficult to explain the large effect (14,000-fold reduction) on the cleavage rate by the simultaneous loss of hydroxyl group and magnesium ion on this basis, because if the former is the only general acid then when it is removed loss of the metal ion should not lower the catalytic efficiency any further. An alternative explanation is therefore that each ribozyme can use either catalytic channel, but under normal circumstances one channel is used predominantly. But when that channel is disrupted then catalysis can still proceed via the second channel. In that case removal of both the key hydroxyl group and the divalent metal ion will prevent both channels operating and the observed rate loss should be much greater.

In summary, the crystal structure of the pseudoknot-containing hammerhead ribozyme[19] shows that its overall structure is strikingly similar to that of the pistol ribozyme. The structure is very compact, and likely very stable, and the PK hammerhead ribozyme is an order of magnitude faster than other hammerhead forms. Both kinetic analysis and the crystallographic data indicate that the 2′-hydroxyl group of G8 is the general acid in the catalytic mechanism, and thus despite the similarity of the overall RNA architecture with the pistol ribozyme, it remains mechanistically a hammerhead ribozyme. Nevertheless, it seems quite likely that it can operate two catalytic channels, although under optimal conditions that using the G8 2′-hydroxyl group predominates.

fast, being more than an order of magnitude faster than the previously-studied extended hammerhead ribozyme under the same conditions[23]. The hammerhead ribozyme has long been recognized as a potential therapeutic RNA species[32–34]. The more robust PK hammerhead ribozyme, with its stable pseudoknot structure, may have enhanced utility within cells, serving as a probe, reagent, or potentially as a therapeutic agent (manuscript in preparation).

Despite the closely similar architecture of the PK hammerhead and pistol ribozymes, our data indicate that they employ distinct catalytic mechanisms. In previous studies comparing the extended hammerhead and pistol ribozymes[23] we concluded that the hammerhead employed the 2′-hydroxyl group of G8 to donate a proton to the 5′-oxyanion leaving group, whereas the pistol ribozyme uses a water molecule bound in the inner-sphere of hydration to protonate the leaving group. In particular the effect of atomic replacement of the O2′ of the hammerhead ribozyme by an amino group on the pH dependence of cleavage was consistent with a direct role in proton transfer, while the response of the analogous change in the pistol ribozyme was completely different[23]. In the current work the sensitivities of cleavage rates to removal of the O2′ groups and magnesium ions in solution were mutually opposite for the PK hammerhead and pistol ribozymes. While the PK hammerhead was more sensitive to removing the 2-hydroxyl group from G8 than loss of magnesium ions, for the pistol ribozyme the sensitivity was reversed, so that the absence of magnesium ions had a greater effect on cleavage rate than the G32 O2′H substitution (Table 1). This is consistent with the general acid being G8 2′OH for the PK hammerhead, and metal ion-bound water for the pistol

## Methods

### Chemical synthesis of RNA oligonucleotides

The RNA oligonucleotides used for crystallization were made by solid-phase chemical synthesis implemented on an Applied Biosystems 394 oligonucleotide synthesizer using *t*-butyldimethyl-silyl chemistry on a 1 μmol scale. Oligoribonucleotides containing 5-bromocytidine were deprotected by incubation with 2 mL of anhydrous 2 M ammonia in methanol for 36 h. All other oligoribonucleotides were deprotected by incubation with 2 mL of a 1:1 (v/v) ammonia:methylamine solution at 60 °C for 20 min. Oligoribonucleotides were then dried under vacuum and dissolved in 100 μl of anhydrous DMSO, 125 μl of triethylamine trihydrofluoride and the mixture was shaken at 65 °C for 2.5 h. After cooling to room temperature the deprotected oligonucleotides were precipitated by adding 25 μl of 3 M sodium acetate and 1 mL of *t*-butyl alcohol and storage at −20 °C. The RNA products were washed twice with anhydrous ethanol and then further purified by electrophoresis under denaturing conditions in 15% polyacrylamide gel containing 7 M urea, extracted from gel fragments by crush-and-soak in 10 mM Tris (pH 8), 1 mM EDTA, 300 mM sodium acetate, and precipitated with ethanol.

### In vitro transcription of hammerhead ribozyme RNA

Hammerhead ribozyme RNA was prepared by in vitro transcription from PCR amplified DNA template by T7 RNA polymerase. A typical in vitro transcription reaction included 0.2 μM dsDNA template, 8 mM NTPs, 50 mM Tris-HCl (pH 7.5), 15 mM MgCl₂, 5 mM DTT, 2 mM spermine, and 40 μl T7 RNA polymerase (~2 mg/ml) in a final volume of 1 mL and incubated for three hours at 37 °C. The final product of in vitro transcription was purified by VAHTS RNA Clean Beads (Vazyme, N412-02, China).

### Crystallization, data collection, and structure determination

The sequences used for crystallization were (5′ to 3′):

ribozyme: GCGAGACAACCAGGAGUCUAUAAAAUUCCCACUGAUGAGACUGGACGAAAGACAUGUAU

substrate: ACAUGUCUdCUGGGA

Substrate 1 with 5-bromocytosine (ᴮʳC): AᴮʳCAUGUCUdCUGGGA

Substrate 2 with ᴮʳC: ACAUGUᴮʳCUdCUGGGA

The two-piece ribozyme RNA molecules for crystallization were prepared by mixing the ribozyme strand (made by transcription) and substrate (chemically synthesized) strand in a 1:1.2 molar ratio in 5 mM HEPES (pH 7), 20 mM NaCl. The RNA mixture was heated to 65 °C and slowly cooled at 4 °C, and the final concentration of the mixture was 100 μM. Each drop was prepared by mixing 1.2 μl of the above RNA mixture with 1.2 μl solutions containing 10 mM magnesium acetate tetrahydrate, 50 mM MES monohydrate (pH 5.6), and 2.5 M ammonium sulfate. The crystals were grown at 10 °C by using the hanging-drop vapor-diffusion method. The crystals appeared after 2-4 days and were flash-frozen in liquid nitrogen. The data were collected either on beamline BL17U (BL02U1) at Shanghai Synchrotron Radiation Facility, China, or at Diamond Light Source in the UK, and the crystals' diffraction extended to 2.89 Å. The initial model was generated using autosol.phenix[36] by the SAD (single anomalous dispersion) method using the anomalous scattering of bromine. The final model was then manually built and subjected to several rounds of adjustment and optimization using Coot and phenix.refine in PHENIX[37]. The statistics are shown in Supplementary Table 1. The structural models were generated by PyMOL2 (http://www.pymol.org).

### Kinetic analysis of hammerhead ribozyme cleavage

Hammerhead ribozyme cleavage kinetics were studied under single-turnover conditions as described previously[23]. Ribozymes were assembled from ribozyme and substrate oligonucleotides:

WT ribozyme: AGACAACCAGGAGUCUAUAAAAUUCCCACUGAUGAGACUGGACGAAAGACAUGU

dG8 ribozyme: AGACAACCAGGAGUCUAUAAAAUUCCCACUGAUdGAGACUGGACGAAAGACAUGU

substrate: ACAUGUCUCUGGGAU

The final reaction contained 1 μM ribozyme and 20 nM radioactively [5′–³²P]-labelled substrate. Standard reaction conditions were 50 mM TAPS (pH 8.0), 1.0 mM MgCl₂, 2.0 M NaCl, 0.1 mM EDTA at 25 °C. Substrate and product were separated by electrophoresis in denaturing 20% polyacrylamide gels and quantified by phosphorimaging, with cleavage rates determined by fitting reaction progress curves to single exponential functions using Kalaidagraph (Abelbeck Software).

### Reporting summary

Further information on research design is available in the Nature Portfolio Reporting Summary linked to this article.

## Data availability

The data supporting the findings of this study are available from the corresponding authors upon request. The crystallographic data generated in this study have been deposited in the PDB database under accession code 8YDC (DOI code to be assigned). The data for our previously-described Pistol ribozyme structure was deposited in the PDB database under accession code 6R47. The raw data used in our ribozyme kinetic analysis is given in the Source data file. Source data are provided with this paper.

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

## Acknowledgements

Research at Nankai University was supported by the National Natural Science Foundation of China (21877065 and 82111530210 to Y.L.); Fundamental Research Funds for the Central Universities, Nankai University (035-63191735 and 035-63221328 to Y.L.); and the Key Project of Tianjin Municipal Natural Science Foundation of China (21JCZDJC00260 to Y.L.). Research in Dundee was supported by a grant from the Engineering and Physical Sciences Research Council (EP/X01567X/1). We thank the Shanghai Synchrotron Radiation Facility (SSRF), Shanghai, China on the BL02U1 (previously BL17U) beamline and Diamond Light Source, UK for synchrotron time.

## Author contributions

Z.L., J.Z. and T.J.W. prepared and validated RNA oligonucleotides, X.Z. carried out crystallography. Y.L. and Y.Y. collected the X-ray diffraction data set and solved the crystal structure. Y.L., T.J.W. and D.M.J.L. analyzed the structure and mechanism and T.J.W. performed the kinetic analysis of wild-type and variant hammerhead ribozymes. Y.L. and D.M.J.L. wrote the manuscript.

## Competing interests

The authors declare no competing interests
