## [Peer Review File · Nature Communications]

The structure and catalytic mechanism of a pseudoknot-containing hammerhead ribozymeREVIEWER COMMENTS

Reviewer #1 (Remarks to the Author):

In this paper, the authors report a crystal structure of a hammerhead ribozyme containing a pseudoknot. Using a 2' H in place of 2' OH at the base adjacent to the scissile phosphate, the authors solved the pre-catalytic structure of the ribozyme with 2.89 Å and 2.79 Å resolution. They note that the global structure of this pseudoknot-containing hammerhead (PK hammerhead) is similar to the pistol ribozyme, with similar stacking of and connectivity between the ribozymes' helices. The cores of the two ribozymes differ, however, as the reported PK hammerhead's active site resembles that of the other hammerhead ribozymes, not the pistol ribozyme's active site. To examine the catalytic mechanism of the PK hammerhead ribozyme, the authors tested its dependence on acid catalysts, Mg²⁺ or the 2' OH of an active site residue previously associated with acid catalysis, as the pistol ribozyme is dependent on a H₂O coordinated to Mg²⁺ and other hammerhead ribozymes use an active site 2' OH. Because the PK hammerhead ribozyme's activity is decreased more by the absence of a 2' OH than the absence of Mg²⁺, unlike the pistol ribozyme, the authors claim that the PK ribozyme uses the same mechanism as other hammerhead ribozymes.

Issues (order of approximately most important to least important):

1. There are no figures comparing the global structures of the hammerhead and pistol ribozymes (side-by-side structures like in Fig.2, for example), despite it being a key part of the paper.
2. There is not much mechanism work; they could include more experiments previously done on the hammerhead and/or pistol ribozymes that have been used to provide insight into their mechanisms.
3. They should be clearer in the abstract that it is similar to pistol in the global structure, not active site structure.
4. They discuss G32 in the pistol ribozyme, but most of the papers on the pistol ribozyme have A at position 32 instead of G. Also, describing it as the base involved in acid catalysis does not seem entirely accurate, because G33 is also key to positioning the Mg²⁺ in pistol.
5. It would be beneficial for Fig. 5 to also show the cleavage site for pistol for comparison instead of just the loop in figure 4 to better show the different active site structures and support that they use different mechanisms.
6. They could directly compare active site structures of the PK hammerhead and extended hammerhead ribozymes to show if there are any noticeable differences that lead to the increase in activity seen in the PK hammerhead.
7. Fig. 2B is only partially color-coded the same as Fig. 2A, so they should match to be clearer.
8. In Fig. 3B, the "An strand" label is not very clear, because the paper does not include the term "An" or "An".
9. Fig. 3A and 3B would be clearer if some or all bases were labeled.
10. The last sentence of the abstract is missing a period.

Reviewer #2 (Remarks to the Author):

The manuscript of Zhan and coworkers presents a crystal structure of the pseudoknot-containing hammerhead ribozyme (PK hammerhead) together with biochemical data to establish whether this ribozyme is more similar to the extended hammerhead or to another related ribozyme, the pistol ribozyme. This is an interesting question that will help understand better ribozyme catalysis in general and the evolution of hammerhead ribozymes in particular. The combination of structural work and the few but crucial biochemical experiments provide the information needed to understand better the PK hammerhead ribozyme. The experimental work is sound and supports well the conclusions. Nevertheless, there are a few comments that need to be addressed.

1. The crystallographic work is sound, but it is surprising that the authors deem the data from the 5Br-containing crystals as better (page 4.). The statistics shown in Supplementary Table I would argue that the derivative data do not extend to the resolution they quote. The mean $I/\sigma I$ and $CC_{1/2}$ in the highest resolution shell is truly marginal. Maybe they want to compare their data at the resolution where the $I/\sigma I$ are the same. I suspect that the resolution of the derivative data may not be as high and probably worse than the native.
2. It would be useful to have a supplementary figure showing more extended regions covered by electron density. Figure 5 shows a very limited region, a larger region would be more informative.

In fact, supplying an electron density map and the observations would be helpful in the review process.

3. The authors compare the structures of PH hammerhead to the pistol ribozymes and show that they are similar. I am surprised that they do not include in their comparisons the extended hammerhead ribozyme. It appears that the PK and canonical hammerhead ribozymes have a similar active site region, but the region around the pseudoknot is very different. It would be informative to add the extended ribozyme to the comparison, including figures of the superposition.

4. Figure 5 shows the core of the ribozymes to emphasize the differences and similarities. I think that having a figure, probably in the supplementary information section, with a superposition of the core of the PK, pistol, and extended hammerhead ribozymes would help to visualize better the descriptions in the text and the structural differences.

5. The authors claim that the extended conformation around the cleavage site is closely similar to the one of the extended hammerhead ribozyme. Closely similar is not a very good way of quantifying similarity. The RMS deviation for the superposed atoms would be more appropriate to quantify the similarity as well as showing a figure in the Supplementary Information section.

6. What are the main differences between the PK and extended hammerhead ribozymes? A comparison similar to the one done for the pistol ribozyme would be appropriate.

7. The authors measured cleavage rates under single-turnover condition and compare them to those of the pistol and extended hammerhead ribozyme. It is confusing from Table 1 how the 14,000 difference is obtained. It should be stated somewhere that the comparisons are against the MgCl₂ + NaCl conditions in all cases.

8. The Table shows the rates for the Extended hammerhead and Pistol ribozymes, but it is not clear where all these rates, like the rates with NaCl only, came from in reference 23. In addition, errors for the rates for the Pistol and extended ribozymes should be included for a proper comparison.

9. Based on the biochemical data and on their previous characterization of the Pistol ribozyme (reference 23), a main conclusion of the work is that there are two catalytic channels (to use the authors' terminology) employed both by the PK and extended hammerhead ribozymes and the pistol ribozyme, but with different preferences. Based on this, the authors conclude that despite the structural similarities between the PK hammerhead and pistol ribozyme, the PK hammerhead is a hammerhead ribozyme. While I do not disagree with this characterization, the authors may wish to consider an alternative explanation: the PK, pistol, and extended ribozymes are all structurally related and share a common two-channel catalytic mechanism, and hence all belong to the same family. As pointed out before in reference 23, they have different catalytic preferences, the pistol ribozyme evolved to employ one catalytic channel while the PK and extended hammerhead evolved to prefer the second one. They are basically the same ribozyme, catalyzing the same reaction using the same structural scaffold.

X. Zhan, T. J. Wilson, Z. Li, J. Zhang, Y. Yang, D. M. J. Lilley and Y. Liu The structure and catalytic mechanism of a pseudoknot-containing hammerhead ribozyme

Responses to reviewers comments

We thank both reviewers for their careful reading, and helpful comments, Most of the comments were questions of clarification, and in most cases the amendment of figures or provision of new figures. These have been itemized at the bottom.

Reviewers comments in blue, responses in black.

To answer each review point by point :

Reviewer #1 (Remarks to the Author):

In this paper, the authors report a crystal structure of a hammerhead ribozyme containing a pseudoknot. Using a 2' H in place of 2' OH at the base adjacent to the scissile phosphate, the authors solved the pre-catalytic structure of the ribozyme with 2.89 Å and 2.79 Å resolution. They note that the global structure of this pseudoknot-containing hammerhead (PK hammerhead) is similar to the pistol ribozyme, with similar stacking of and connectivity between the ribozymes' helices. The cores of the two ribozymes differ, however, as the reported PK hammerhead's active site resembles that of the other hammerhead ribozymes, not the pistol ribozyme's active site. To examine the catalytic mechanism of the PK hammerhead ribozyme, the authors tested its dependence on acid catalysts, Mg²⁺ or the 2' OH of an active site residue previously associated with acid catalysis, as the pistol ribozyme is dependent on a H₂O coordinated to Mg²⁺ and other hammerhead ribozymes use an active site 2' OH. Because the PK hammerhead ribozyme's activity is decreased more by the absence of a 2' OH than the absence of Mg²⁺, unlike the pistol ribozyme, the authors claim that the PK ribozyme uses the same mechanism as other hammerhead ribozymes.

Issues (order of approximately most important to least important):

1. There are no figures comparing the global structures of the hammerhead and pistol ribozymes (side-by-side structures like in Fig.2, for example), despite it being a key part of the paper.

We did in fact show a superposition of the two structures in Supplementary Figure S2. However we have added a side-by-side comparison of the two ribozymes in the same figure (now Supplementary Figure S3 A, B) as requested.

2. There is not much mechanism work; they could include more experiments previously done on the hammerhead and/or pistol ribozymes that have been used to provide insight into their mechanisms.

In this paper the important data differentiating the hammerhead and pistol mechanisms are two-fold. These are the data showing that the hammerhead is more sensitive to removal of the key O2' group than to the absence of Mg²⁺ ions, which is the reverse of case for the pistol ribozyme. In addition, the structural data point to the same conclusion; the O2'-O5' distance is 2.9 Å in the PK hammerhead structure, whereas it is 4.7 Å in the pistol. It is really the combination of these two observations (biochemical and structural) that gives us the confidence to argue for the mechanistic difference between the two ribozymes. The original key experiment differentiating the two ribozyme mechanisms (presented in our 2019 paper) was the effect of 2'-amino-ribose substitutions. Ideally it would be good to repeat this for the PK hammerhead ribozyme. Unfortunately our collaborators no longer have stocks of that phosphoramidite. But we feel that even in the absence of such data those that we present are compelling.

3. They should be clearer in the abstract that it is similar to pistol in the global structure, not active site structure.

We do say “The overall structure of the PK hammerhead is closely similar to that of the pistol ribozyme”. We have now added “Despite the similarity in overall structure to the pistol ribozyme, the local structure close to the cleavage site differs, and the PK hammerhead retains its unique mechanistic identity ...”.

4. They discuss G32 in the pistol ribozyme, but most of the papers on the pistol ribozyme have A at position 32 instead of G. Also, describing it as the base involved in acid catalysis does not seem entirely accurate, because G33 is also key to positioning the Mg²⁺ in pistol.

Position 32 can be G or A, with a 60:40 ratio. The interactions are made with the O2' and the N7, so these are both possible with either purine. We originally (2019) determined the crystal structure with G32, but all our kinetics have been done with A32. This gives a rate of cleavage of $9.8 \pm 0.6 \text{ min}^{-1}$. Our measured rate with G32 is $7.3 \pm 0.6 \text{ min}^{-1}$. These data were published in the 2019 paper. Thus both forms are substantially active.

As regards the positioning of the catalytic Mg²⁺ ion in the pistol ribozyme, a number of contacts are important. It is held by interactions with G33 N7 (a direct metal-nitrogen bond), with the G32 O2', and with a non-bridging O of the scissile phosphate. This is not discussed in the text.

5. It would be beneficial for Fig. 5 to also show the cleavage site for pistol for comparison instead of just the loop in figure 4 to better show the different active site structures and support that they use different mechanisms.

This is a good idea. We have done this, with the corresponding view of the cleavage site of the pistol ribozyme shown as Figure 5 part B.

6. They could directly compare active site structures of the PK hammerhead and extended hammerhead ribozymes to show if there are any noticeable differences that lead to the increase in activity seen in the PK hammerhead.

This is a good point, made by both reviewers. We had originally made a comparison in an earlier draft, and then removed it for clarity. We have now added a comparison of the structures of our new PK hammerhead ribozyme with that of the Martick and Scott extended as Supplementary Figure S4.

7. Fig. 2B is only partially color-coded the same as Fig. 2A, so they should match to be clearer.

It was not easy to get the color right in Illustrator. However, we have now succeeded to make the cylinder representing helix I a more pink shade.

8. In Fig. 3B, the “An strand” label is not very clear, because the paper does not include the term “An” or “An”.

The sequence is actually AUA₄ so we have changed Figure 3B accordingly. An was just a short-hand form.

9. Fig. 3A and 3B would be clearer if some or all bases were labeled.

It becomes too cluttered if we try to label every nucleotide, but we have added labels to some, allowing the reader to work out the identity of each nucleotide.

10. The last sentence of the abstract is missing a period.

We have restored the delinquent full stop.

Reviewer #2 (Remarks to the Author):

The manuscript of Zhan and coworkers presents a crystal structure of the pseudoknot-containing hammerhead ribozyme (PK hammerhead) together with biochemical data to establish whether this ribozyme is more similar to the extended hammerhead or to another related ribozyme, the pistol ribozyme. This is an interesting question that will help understand better ribozyme catalysis in general and the evolution of hammerhead ribozymes in particular. The combination of structural work and the few but crucial biochemical experiments provide the information needed to understand better the PK hammerhead ribozyme. The experimental work is sound and supports well the conclusions. Nevertheless, there are a few comments that need to be addressed.

1. The crystallographic work is sound, but it is surprising that the authors deem the data from the 5Br-containing crystals as better (page 4.). The statistics shown in Supplementary Table I would argue that the derivative data do not extend to the resolution they quote. The mean I/σ and $CC_{1/2}$ in the highest resolution shell is truly marginal. Maybe they want to compare their data at the resolution where the I/σ are the same. I suspect that the resolution of the derivative data may not be as high and probably worse than the native

We completely agree with this comment. Although the extremely extended resolution cut of the dataset did not affect the generation of the initial model for SAD solution, we acknowledge the necessity for enhanced data processing to guarantee a high-quality publication. We have reprocessed the dataset containing 5Br with a resolution cut at 3.5 Å, resulting in mean I/σ and $CC_{1/2}$ values of 22.87 (3.14) and 1 (0.994) respectively. Furthermore, we re-ran the AutoSol program in the PHENIX suite and obtained the same correct solution as before.

Electron density map of PK hammerhead ribozyme solved and processed by AutoSol in PHENIX. The map is contoured at 2.5σ . The subsequent AutoBuild process modeled 62 nucleotides into the density map.

2. It would be useful to have a supplementary figure showing more extended regions covered by electron density. Figure 5 shows a very limited region, a larger region would be more informative. In fact, supplying an electron density map and the observations would be helpful in the review process.

Electron density was also shown in Figure 3 C,D, so density was shown for three different sections of the molecule. But we take the point, and we have now added a stereo image of the entire PK hammerhead ribozyme with its electron density in the SI as Supplementary Figure S1. In addition, we thought it would be valuable to add an image of the PK hammerhead loop with the electron density for the important G8 shown, to show there is no ambiguity in the position of the nucleobase. This is shown as the additional Supplementary Figure S6.

3. The authors compare the structures of PH hammerhead to the pistol ribozymes and show that they are similar. I am surprised that they do not include in their comparisons the extended hammerhead ribozyme. It appears that the PK and canonical hammerhead ribozymes have a similar active site region, but the region around the pseudoknot is very different. It would be informative to add the extended ribozyme to the comparison, including figures of the superposition.

This is a good point. We had originally made a comparison in an earlier draft, and then removed it for clarity. We have now added a comparison of the structures of our new PK hammerhead ribozyme with that of the Martick and Scott extended as Supplementary Figure S2.

4. Figure 5 shows the core of the ribozymes to emphasize the differences and similarities. I think that having a figure, probably in the supplementary information section, with a superposition of the core of the PK, pistol, and extended hammerhead ribozymes would help to visualize better the descriptions in the text and the structural differences.

As shown here, the catalytic core of the hammerhead and pistol ribozymes are structurally distinct. Therefore a superposition failed to give a good alignment. So we feel it has no value to show this in the paper.

5. The authors claim that the extended conformation around the cleavage site is closely similar to the one of the extended hammerhead ribozyme. Closely similar is not a very good way of quantifying similarity. The RMS deviation for the superposed atoms would be more appropriate to quantify the similarity as well as showing a figure in the Supplementary Information section.

As shown in Supplementary Figure S3, the loops of the PK and extended hammerhead ribozymes are very closely similar. By superimposing the catalytic cores of Sm α -HHRz or sTRSV-HHRz with the PK hammerhead, we obtain RMSD values of 1.285 and 0.292 Å, respectively. This is now noted in the text.

6. What are the main differences between the PK and extended hammerhead ribozymes ?

In short the global structures between the PK and extended hammerhead ribozymes is the presence of the pseudoknot, as shown in Supplementary Figure S2, but the loop structures are virtually identical, as shown in Supplementary Figure S3.

7. The authors measured cleavage rates under single-turnover condition and compare them to those of the pistol and extended hammerhead ribozyme. It is confusing from Table 1 how the 14,000 difference is obtained. It should be stated somewhere that the comparisons are against the MgCl₂ + NaCl conditions in all cases.

All rates are referenced to those of the wild type sequence with the 2'-hydroxyl present at the 8 (hammerhead) or 32 (pistol) position, and in the presence of both sodium and magnesium ions. This has been made clearer in the text.

8. The Table shows the rates for the Extended hammerhead and Pistol ribozymes, but it is not clear where all these rates, like the rates with NaCl only, came from in reference

23. In addition, errors for the rates for the Pistol and extended ribozymes should be included for a proper comparison.

All PK hammerhead data newly obtained in this work. The majority of the pistol results were taken from our previous paper (ref. 23), but a few were measured in the present work. This has been clarified in the legend to **Table 1**. We have added standard deviations for all rates in **Table 1**, including those measured previously.

9. Based on the biochemical data and on their previous characterization of the Pistol ribozyme (reference 23), a main conclusion of the work is that there are two catalytic channels (to use the authors' terminology) employed both by the PK and extended hammerhead ribozymes and the pistol ribozyme, but with different preferences. Based on this, the authors conclude that despite the structural similarities between the PK hammerhead and pistol ribozyme, the PK hammerhead is a hammerhead ribozyme. While I do not disagree with this characterization, the authors may wish to consider an alternative explanation: the PK, pistol, and extended ribozymes are all structurally related and share a common two-channel catalytic mechanism, and hence all belong to the same family. As pointed out before in reference 23, they have different catalytic preferences, the pistol ribozyme evolved to employ one catalytic channel while the PK and extended hammerhead evolved to prefer the second one. They are basically the same ribozyme, catalyzing the same reaction using the same structural scaffold.

This is broadly what we were saying, but would hesitate to state it with certainty. The structural data (particularly the structure of the loops and consequently the O2'-O5' distances) are clearly distinct for the two ribozymes, despite the close similarity of the overall architecture. But the major losses of activity seen on the simultaneous removal of the O2' and Mg²⁺ ions would be consistent with both ribozymes being capable of using either mechanism. But our data suggest there is a strong preference of the hammerheads to use the O2' as general acid, while the pistol uses Mg²⁺-bound water as general acid in the absence of perturbation. I think we are really saying much the same thing as the reviewer. This prompts us think that perhaps a bioinformatic investigation of the phylogenetic origins of the two ribozymes might be informative.

ABSTRACT

This has been revised to take into account reviewers suggestions, to conform to journal style and to conform to journal length (150 words – it was previously over this limit).

SUMMARY OF NEW OR AMENDED FIGURES

Main figures

Fig 2 : helix I in part B recoloured

Fig 3A, B : some nucleotides numbered

Fig 3B : An strand renamed AUA₄

Fig 5 : pistol ribozyme active site added as part B.

Supplementary figures

Supp Fig S1 : additional figure showing whole PKHH with electron density map

Supp Fig S2 : additional figure comparing the extended and PK hammerhead global structures

Supp Fig S3 : additional figure comparing the loops of the extended and PK hammerhead ribozymes

Supp Fig S5 : added global structures of PK hammerhead and pistol side by side (new parts A and B)

Supp Fig S6 : additional figure showing electron density for G8 in the PK hammerhead ribozyme structure

Best regards

David Lilley and Yijin Liu

University of Dundee *Nankai University*

REVIEWERS' COMMENTS

Reviewer #2 (Remarks to the Author):

The authors have answered the concerns appropriately. My only remaining comment is that in the revised Abstract, line 41, it is not clear what they mean ("the structure is almost perfectly in-line ..."). Which structure is perfectly in-line?

There was a single comment from a reviewer :

Reviewer #2 (Remarks to the Author):

The authors have answered the concerns appropriately. My only remaining comment is that in the revised Abstract, line 41, it is not clear what they mean (“the structure is almost perfectly in-line ...”). Which structure is perfectly in-line?

Although the original text would be quite understandable to anyone in the ribozyme field, we take the point that it could be clearer. So we revised the abstract to say “ Here we show that nucleophilic attack is almost perfectly in-line “. The meaning is now completely unambiguous.